# Association of Sugar-Sweetened Beverages and Cardiovascular Diseases Mortality in a Large Young Cohort of Nearly 300,000 Adults (Age 20–39)

**DOI:** 10.3390/nu14132720

**Published:** 2022-06-29

**Authors:** Chien-Hua Chen, Min-Kuang Tsai, June-Han Lee, Christopher Wen, Chi-Pang Wen

**Affiliations:** 1Digestive Disease Center, Changbing Show-Chwan Memorial Hospital, Lukang 50544, Taiwan; showchench@gmail.com; 2Department of Food Science and Technology, Hungkuang University, Taichung 43302, Taiwan; 3Department of Post-Baccalaureate Medicine, College of Medicine, National Chung Hsing University, Taichung 402202, Taiwan; 4Institute of Population Health Sciences, National Health Research Institutes, Zhunan 35053, Taiwan; minlight@nhri.org.tw (M.-K.T.); jhlee@nhri.org.tw (J.-H.L.); 5Long Beach VAMC Hospital, University of Irvine Medical Center, Irvine, CA 90822, USA; cw691434@gmail.com; 6Graduate Institute of Clinical Medical Science, China Medical University, Taichung 40402, Taiwan

**Keywords:** sugar-sweetened beverages, cardiovascular diseases, mortality, younger adults, dose–response relationship

## Abstract

(1) Background: The association of sugar-sweetened beverages (SSBs) with cardiovascular disease (CVD) mortality in younger adults (age 20–39) is rarely mentioned in the literature. Younger adults are less vulnerable to CVDs, but they tend to consume more SSBs. This prospective study aimed to assess the association between CVD mortality and SSBs in younger adults between 1994 and 2017. (2) Methods: The cohort enrolled 288,747 participants consisting of 139,413 men and 148,355 women, with a mean age 30.6 ± 4.8 years, from a health surveillance program. SSBs referred to any drink with real sugar added, such as fructose corn syrup or sucrose. One serving of SSB contains about 150 Kcal of sugar in 12 oz of drink. Cox models were used to estimate the mortality risk. (3) Results: There were 391 deaths from CVDs in the younger adults, and the positive association with CVD mortality started when SSB intake was ≥2 servings/day (HR: 1.59, 95% CI: 1.16–2.17). With mortalities from diabetes and kidney disease added to CVDs, the so-called expanded CVD mortality risk was 1.49 (95% CI: 1.11–2.01). By excluding CVD risk factors (hypertension, diabetes, and smoking), the CVD mortality risk increased to 2.48 (95% CI: 1.33–4.62). The dose–response relationship persisted (*p* < 0.05 for trend) in every model above. (4) Conclusions: Higher intake of SSBs (≥2 servings/day) was associated with increased CVD mortality in younger adults. The younger adults (age 20–39) with SSB intake ≥2 servings/day had a 50% increase in CVD mortality in our study, and the mortality risk increased up to 2.5 times for those without CVD risk factors. The dose–response relationship between the quantity of SSB intake and the mortality risk of CVD in younger adults discourages SSB intake for the prevention of CVD mortality.

## 1. Introduction

Cardiovascular diseases (CVDs), mainly consisting of ischemic heart disease and stroke, are the leading cause of mortality and impaired quality of life worldwide [1,2,3]. Despite the decline in CVD mortality in the elderly since 2003, CVD mortality has steadily increased for adults aged ≤55 [4]. Based on the Framingham Heart Study, the estimated incidence of acute myocardial infarction (AMI) in 2020 for individuals aged 30–34 was about 12.9/1000 for men and 2.2/1000 for women, respectively [5]. On other hand, the estimated incidence of AMI for individuals aged 35–44 was about 38.2/1000 for men and 5.2/1000 for women, respectively [5]. Similarly, the annual incidence of ischemic stroke has substantially increased for adults aged <50 years, with more than two million cases reported worldwide [6,7,8,9,10,11,12]. However, the discussion around the prevention of CVDs in younger adults receives less attention and is challenging because their etiologies and risk factors are different from those of older adults. 

“Sugar-sweetened beverages” (SSBs) are a major source of added sugar for drinks and SSB intake in adults has remained alarmingly high worldwide [13,14,15]. It has been reported that SSB intake could increase CVD incidence [16,17,18,19]. Some studies even supported the association of SSB intake with increased all-cause mortality, and death mainly resulted from CVDs, rather than from malignancy [13,14,20,21,22,23,24,25,26,27,28,29]. Through increasing the sugar supply, it was supposed that SSBs could enhance insulin resistance to increase CVD mortality by inducing the risk factors of CVDs, such as diabetes, hypertension, CAD, obesity, frailty, and arterial stiffness [21,22,30,31,32,33]. However, a report from Singapore could not find the association between CVD mortality and SSB intake based on a Chinese adult cohort [34], and neither did a study among elderly participants from the United States [35]. 

Younger adults are less vulnerable to CVDs, although they tend to consume more SSBs than older adults [14,36,37]. Among Taiwanese adults aged 19–44 years, with a mean of 7.8 servings/week, about 83.6% of them consumed more than 1 serving/week of SSBs between 2013 and 2016 [38]. The consumption of SSBs in the whole Taiwan has increased by 8.9% annually between 2005 and 2019, and SSB intake in the whole of Asia has been expected to increase in the coming decade [39,40,41]. Although many previous studies have investigated the association between SSB intake and CVD mortality by age groups [13,20,21,22,23,24,25,26,27,28,29], the association between SSB intake and CVD mortality or incidence in younger adults (age 20–39) is rarely mentioned, particularly in the Asian population [42,43].

We conducted this prospective study with 288,747 participants aged 20–39 from a self-paid medical screening program to assess the association between CVD mortality and SSB intake among younger adults. In considering the high likelihood of classifying CVD mortality as death from diabetes in patients with diabetes or as death from kidney disease in patients with kidney disease, we also calculated the mortality risks of expanded CVDs as a sensitivity analysis by including the registered mortalities related to CVDs, diabetes, or kidney diseases [44].

## 2. Materials and Methods

### 2.1. Study Population

Based on Encyclopedia Britannica, the definition of younger adults was taken as adults aged between 20 and 39. The MJ Institution owns four clinics across Taiwan that have consecutively enrolled 288,747 younger adults, who could provide complete information about SSB intake, for a self-paying medical screening program since 1994 [44]. We followed up on the participants between 1994 and 2017; the interquartile was 9–20 years with a median period of 15 years (Figure 1). The screening program was a membership system and the participants could pay for repeated physical check-ups and receive a multidisciplinary education for individualized counselling during examinations. 

### 2.2. Laboratories and Questionnaires

Each of the participants received a physical examination and a review of their medical history with self-administered structured questionnaires for lifestyle information prior to a series of standard tests for blood, urine, electrocardiography, chest radiography, and pulmonary function. The physicians would review and ascertain the personal history of the participants during the physical examination. We defined hypertension as those with a known history, systolic blood pressure ≥140 mmHg, diastolic blood pressure ≥90 mmHg, or treated with anti-hypertensive agents. We defined diabetes as those with a known history, fasting blood glucose level ≥126 mg/dL, or treated with hypoglycemic agents for controlling blood sugar. Regular alcohol drinkers were defined as those consuming ≥2 alcoholic drinks/day on three or more days a week, and occasional alcohol drinkers as those consuming less than regular alcohol drinkers [45].

SSBs referred to drinks with real sugar added. The items registered in the self-reported questionnaire pertaining to SSBs included caffeinated or de-caffeinated cola, carbonated beverages, noncarbonated beverages, sports drinks, vitamin-enhanced drinks, cocoa, lemonades, fruit juice concentrates, and fruit punch. We only classified drinks with real sugar added, such as fructose corn syrup or sucrose, as SSBs in this study. Artificially sweetened beverages (ASBs) without real sugar added were not studied here. We classified the amount of SSB intake into four categorizes according to the participants’ answers to self-administered questionnaires about SSB intake in the most recent month: 0 to <0.5 serving/day, ≥0.5 to <1 serving per day, ≥1 to <2 servings per day, and ≥2 servings per day. Each serving contains 12 oz with about 150 Kcal or 35 g of added sugar [22]. Before the leaving of the participants in the afternoon, the dieticians would verify the dietary habits from the answered questionnaires, including the average total daily calorie intake, to provide individualized face-to-face counselling.

The participants reported the types and intensities of weekly leisure time physical activity (LTPA) that they did during the previous month, and intensity was classified into four categories with several examples of exercise types given: light (walking), moderate (brisk walking), medium-vigorous (jogging), or high-vigorous (running). A metabolic equivalent value (MET; 1 MET = 1 kcal per h per kg of bodyweight) was calculated according to the basis of Ainsworth’s compendium of physical activities [46]. We would assign a weighted MET value based on the length of time in each intensity category when the subjects’ activities were more than one category.

The MET value was based on the length of time in each intensity category when the subject’s activities were more than one category. The LTPA volume/week was calculated by the product of MET and duration of exercise (h) per week. The LTPA was classified into five subgroups, including inactive (<3.75 MET h/week or <5 min/day), low active (3.75–7.49 MET h/week or about 15 min/day), medium (7.50–16.49 MET h/week or about 30 min/day), high (16.50–25.49 MET h/week or about 60 min/day), and very high active (≥25.50 MET h/week or about 90 min/day or more). All the data pertaining to physical activities have been addressed and validated in our former report [45].

### 2.3. Assessment of Outcome

The registered cause-specific mortalities between 1994 and 2017 were matched based on the National Death File [47]. The International Classification of Diseases, 9th version (ICD-9), was used to classify the major cause-specific mortality, consisting of CVDs (390–459) and expanded CVDs with CVDs as well as diabetes (250) and kidney diseases (580–589). The mortalities from ischemic heart disease (410–414) and stroke (430–434 and 436) were also analyzed in this study. We compared the mortality risks of CVDs, ischemic heart disease, stroke, and expanded CVDs between subjects consuming different amounts of SSBs. Each participant provided consent and the Institutional Review Board of China Medical University approved this study. We encrypted all the data during the entire study process.

### 2.4. Statistical Analysis

We measured the hazard ratios (HRs) with 95% confidence intervals of mortality by adjusting age, sex, education levels, smoking status, alcohol drinking status, physical activity, body mass index (BMI), hypertension, and diabetes with Cox models. We conducted the following sensitivity analyses with the respective aims to validate our findings: (1) To evaluate the dose–response relationship by measuring a *p*-value for the trend between SSB intake and the mortality risks of CVDs; (2) To examine the association between SSB intake and CVD mortality risk by stratified analyses based on sex, smoking, hypertension, diabetes, BMI, and daily calorie intake of body weight, excluding those died within 3 years after enrollment, and excluding regular alcohol drinkers; (3) To evaluate the independent role of SSBs in contributing to CVD mortality by excluding those with CVD-related risk factors, such as smoking, hypertension, or diabetes; (4) To evaluate the mortality risk of CV-related diseases by measuring the mortality of expanded CVDs with diabetes and kidney diseases added [26]. Except for the stratification criteria, HR was adjusted for categories of age, sex, education levels, smoking status, alcohol drinking status, physical activity, body mass index, hypertension, and diabetes in performing the sensitivity analysis. With no violation of the proportional hazard assumption, our study used SAS 9.4 (SAS Institute Inc., Cary, NC, USA) for statistical analyses and considered it statistically significant when a two-tailed *p* was <0.05.

## 3. Results

### 3.1. Population Distribution of SSB Consumption

Overall, the study cohort enrolled 288,747 younger adults with a mean age of 30.6 ± 4.8 years, including 139,413 men and 148,355 women (Table 1). Our findings showed that those who were men, current smokers, regular alcohol drinkers, had hypertension, or had diabetes tended to have SSB intake greater than 2 servings/day. 

### 3.2. SSB Consumption and Mortality Risks of CVD

Table 2 shows that there were 391 deaths of CVD in the cohort. The mortality from CVDs (HR: 1.59, 95% CI: 1.16–2.17) or expanded CVDs (HR: 1.49, 95% CI: 1.11–2.01) was positively correlated to SSB intake ≥2 serving/day, with a dose–response relationship for any amount (*p* < 0.05 for trend). 

### 3.3. Stratified Analyses for Association of SSB Intake with CVD Mortality

Table 3 shows that the association of SSB intake ≥2 servings/day with mortality from CVDs among younger adults was significant for men (HR: 1.48, 95% CI: 1.03–2.12), women (HR: 1.90, 95% CI: 1.01–3.61), non-smokers (HR: 1.95, 95% CI: 1.14–3.32), BMI < 30 kg/m^2^ (HR: 1.54, 95% CI: 1.08–2.19), those with non-hypertension (HR: 1.89, 95% CI: 1.30–2.73), non-diabetes individuals (HR: 1.54, 95% CI: 1.11–2.14), those with daily calorie intake ≥25 Kcal/kg body weight (HR: 1.87, 95% CI: 1.19–2.95), and those with daily calorie intake <25 Kcal/kg body weight (HR: 1.75, 95% CI: 1.04–2.93); those who died in 3 years were excluded (HR: 1.66, 95% CI: 1.20–2.29), as well as those without regular alcohol drinking (HR: 1.58, 95% CI: 1.16–2.17). It was noted that the dose–response relationship was consistently present in the above stratified analyses, except for women and those with daily calorie intake <25 Kcal/kg body weight. The aforementioned association with CVDs (HR: 2.48, 95% CI: 1.33–4.62) even persisted in CVD risk-free younger adults after excluding the subjects with smoking, hypertension, or diabetes, similarly with *p* < 0.05 for trend. However, the positive association of SSB intake ≥2 servings/day with mortality from CVD could not be found in ever smokers, BMI ≥ 30 kg/m^2^, hypertension, or diabetes. 

## 4. Discussion

In this study, we found that SSB intake was associated with CVD mortality in younger adults. The mortality risk persisted even after adjustment or exclusion of the possible confounders such as smoking, alcohol drinking, hypertension, diabetes, and obesity. A dose–response relationship was found between SSB intake and CVD mortality for younger adults, aged <40, with those consuming ≥2 servings/day showing significantly increased mortality risk from CVDs. SSB intake was also positively associated with the mortality risk from expanded CVDs, with the inclusion of mortalities related to diabetes or kidney diseases.

With a dose–response relationship, the positive association of SSB intake ≥2 servings/day with mortality from CVDs for non-smokers, non-obesity (BMI < 30 kg/m^2^), non-hypertension, and no-diabetes might portend an independent role of SSB in contributing to CVD mortality. However, SSB might be less important than smoking, obesity, hypertension, and diabetes in contributing to CVD mortality because the aforementioned association was not statistically significant in the presence of the above risk factors of CVDs. Owing to the relatively rare outcomes of CVD mortality in younger adults, some studies could only suggest that SSB may indirectly increase the CVD risk by enhancing the development of the cardiometabolic risk factors, such as obesity and diabetes [48,49,50]. Thus, based on our findings, more studies are required to clarify the mechanisms for the association of SSB with CVD mortality, either an independent risk factor or an epiphenomenon.

Through enrolling a large number of participants, it was noted that younger adults were vulnerable to CVDs when they drank more than 2 servings/day of SSBs. As a matter of fact, SSB intake in Taiwan has dramatically risen 8.9% every year in the last 15 years [14]. It is reasonable to project that CVD mortality of younger adults may emerge in Taiwan if they start to drink more like those in United States, with 1/2 of adults and 2/3 of younger adults (age <40) consuming ≥1 serving/day [51]. We found that the mortality risks of diabetes and kidney diseases for the younger adults were increased (HR > 1) after SSB intake ≥1 serving/day, although there was no statistical significance (Table 2). It is highly possible for the mortality risks to increase significantly if the population and SSB intake increase. The increasing trend of CVD mortality in younger adults could not be fully explained by the habitus of smoking or alcohol drinking because these behaviors have been highly prevalent for decades before the prevalence of CVD became evident. In contrast, the lag effect of 10–20 years between SSB intake and CVD mortality for younger adults (enrolled mean age: 30.6 ± 4.8 years), with a mean age 45 ± 7.6 years for the development of CVD mortality, was similar to the temporal association of tobacco with cancer mortality [52].

It has been reported that high SSB intake was frequently accompanied by an unhealthy lifestyle of smoking and comorbidities of obesity or diabetes [21,22,23]. Consistent with the literature, we found that either current smokers or regular alcohol drinkers tended to have SSB consumption greater than 2 servings/day. More studies are required to clarify whether heathier lifestyle modification without smoking or alcohol drinking can simultaneously diminish SSB intake. However, our study supports the rationale for discouraging SSB intake, smoking, and alcohol drinking to avert the mortality risk of CVD, particularly for younger adults. Although higher SSB (≥2 servings/day) consumers tended to have hypertension or diabetes, the CVD mortality possibly attributable to SSB persisted in our sensitivity analyses.

There were a number of strengths in this study. First, the well-categorized SSB intake provided an objective assessment of CVD mortality risk based on the quantity of SSB intake. Second, the structured-questionnaires were answered at the beginning of enrollment; therefore, recall bias could be mitigated. Third, we excluded participants who were dead within 3 years of follow-up to avoid incipient comorbidities or competing factors. Fourth, the accuracy for our registration of cause-specific incidence and mortality was high [53,54]. 

There are several limitations in this study. First, the participants were selected from a paid health program, which may imply a significant socio-economic bias. The MJ clinics provide incentives, paid for by the head of the household, to conduct a family-centered screening program by recruiting extended family members such as uncles, cousins, or grandparents. With a large sample size (288,747 younger adults) and long follow-up period (median: 15 years), the MJ database has shown high concordance with the Taiwan population in the prevalence of cancers and common cancer-related risk factors, such as smoking and alcohol drinking [40]. The socio-economic status bias could be minimized by internal comparison of the relative mortality risks in this study, although our results required external validation. Second, causality can never be claimed based on a single observational study. We might not consider some residual confounding factors in this study, but the impact of high SSB intake on the mortality risk from CVD for younger adults persisted in the multivariable analysis and all the sensitivity analyses. Third, the habits of SSB intake and other CVD-associated risk factors might change with time, which would influence our findings. Our study showed an acceptable correlation of SSB intake in those participants with two visits (time interval: 2.3 ± 2.08 years), with a value of 0.41 for the Spearman’s rank correlation coefficient (Table 4). Fourth, with the lowest ASB utilization (less than 1 mL per person per day) for the entire Asia-Pacific area between 2000 and 2014, we could not evaluate the impact of ASB on the mortality risk, as most sweetened beverages in Taiwan contained added real sugar [55]. 

In conclusion, younger adults (age 20–39) with SSB intake ≥2 servings/day displayed a 50% increase in CVD mortality and the mortality risks increased up to 2.5 times for those without CVD risk factors. The dose–response relationship between the quantity of SSB intake and the mortality risk of CVD for younger adults warrants the discouraging of SSB intake for the prevention of CVD. More studies are required to ascertain the causality and the external validity. 

## Figures and Tables

**Figure 1 nutrients-14-02720-f001:**
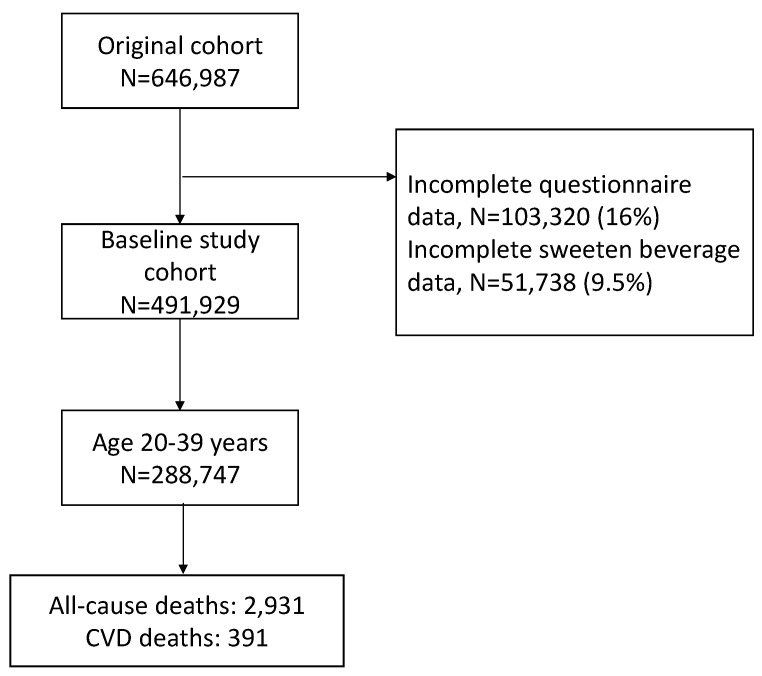
Flowchart of the enrolled participants.

**Table 1 nutrients-14-02720-t001:** The characteristics of the younger participants in this study.

SSB Intake (Serving/Day)		Total (n)	(%)	None (n)	(%)	>0–<0.5 (n)	(%)	≥0.5–<1 (n)	(%)	≥1–<2 (n)	(%)	≥2 (n)	(%)
Age	20–39	288,747	(100.0)	80,930	(28.0)	86,915	(30.1)	49,669	(17.2)	44,388	(15.4)	26,845	(9.3)
Sex	Men	139,413	(100.0)	34,231	(24.6)	40,463	(28.9)	26,425	(19.0)	22,041	(15.8)	16,253	(11.7)
	Women	148,355	(100.0)	46,344	(31.2)	46,175	(31.1)	23,093	(15.6)	22,217	(15.0)	10,526	(7.1)
Education	Middle school or below	9480	(100.0)	4975	(52.5)	1948	(20.5)	734	(7.7)	1038	(10.9)	785	(8.4)
	High school	60,771	(100.0)	21,911	(36.1)	15,664	(25.8)	7981	(13.1)	8929	(14.7)	6286	(10.3)
	Junior college	71,350	(100.0)	20,021	(28.1)	21,143	(29.6)	12,252	(17.2)	11,043	(15.4)	6891	(9.7)
	College or above	142,924	(100.0)	32,670	(22.9)	46,938	(32.8)	28,034	(19.6)	22,762	(15.9)	12,520	(8.8)
Smoking status	Non-smoker	203,091	(100.0)	56,848	(28.0)	65,883	(32.4)	35,779	(17.6)	29,956	(14.8)	14,625	(7.2)
	Ex-smoker	13,810	(100.0)	3849	(27.9)	4072	(29.5)	2444	(17.7)	2084	(15.0)	1361	(9.9)
	Current smoker	64,695	(100.0)	16,574	(25.6)	15,419	(23.8)	10,678	(16.6)	11,599	(17.9)	10,425	(16.1)
Alcohol drinking status	Non-drinker	232,363	(100.0)	61,571	(26.5)	72,487	(31.2)	41,468	(17.8)	36,234	(15.6)	20,603	(8.9)
	Occasional drinker	29,097	(100.0)	9716	(33.3)	7417	(25.5)	4498	(15.5)	4240	(14.6)	3226	(11.1)
	Regular drinker	16,938	(100.0)	5602	(33.1)	4159	(24.6)	2356	(13.9)	2529	(14.9)	2292	(13.5)
Physical activity	Inactive	153,182	(100.0)	42,298	(27.6)	44,174	(28.8)	26,028	(17.0)	24,642	(16.1)	16,040	(10.5)
	Low	79,432	(100.0)	22,999	(29.0)	25,513	(32.1)	13,778	(17.3)	11,396	(14.4)	5746	(7.2)
	Medium	34,026	(100.0)	9162	(26.9)	10,702	(31.5)	6173	(18.1)	5116	(15.0)	2873	(8.5)
	High	9938	(100.0)	3036	(30.5)	2835	(28.5)	1654	(16.6)	1413	(14.3)	1000	(10.1)
	Very high	6637	(100.0)	1847	(27.8)	1777	(26.8)	1174	(17.7)	1021	(15.4)	818	(12.3)
Body mass index	<18.5	36,489	(100.0)	10,524	(28.8)	11,262	(30.9)	6105	(16.7)	5651	(15.5)	2947	(8.1)
	18.5~24	191,429	(100.0)	55,040	(28.8)	57,610	(30.1)	32,620	(17.0)	29,143	(15.2)	17,016	(8.9)
	25~29	48,990	(100.0)	12,346	(25.2)	14,546	(29.7)	8813	(18.0)	7759	(15.8)	5526	(11.3)
	≥30	10,765	(100.0)	2635	(24.5)	3198	(29.7)	1970	(18.3)	1682	(15.6)	1280	(11.9)
Hypertension	None	269,387	(100.0)	75,252	(27.9)	81,293	(30.2)	46,347	(17.2)	41,627	(15.5)	24,868	(9.2)
	Yes	18,381	(100.0)	5323	(29.0)	5345	(29.1)	3171	(17.3)	2631	(14.3)	1911	(10.4)
Diabetes	None	284,574	(100.0)	79,563	(28.0)	85,702	(30.1)	49,042	(17.2)	43,835	(15.4)	26,432	(9.3)
	Yes	3194	(100.0)	1012	(31.7)	936	(29.3)	476	(14.9)	423	(13.2)	347	(10.9)
Chronic kidney disease	None	255,803	(100.0)	70,838	(27.7)	77,335	(30.2)	44,458	(17.4)	39,405	(15.4)	23,767	(9.3)
	Stage 1	7569	(100.0)	2334	(30.8)	2137	(28.2)	1193	(15.8)	1071	(14.2)	834	(11.0)
	Stage 2	5698	(100.0)	1810	(31.8)	1526	(26.8)	897	(15.7)	795	(13.9)	670	(11.8)
	Stage 3	555	(100.0)	185	(33.3)	142	(25.6)	81	(14.6)	84	(15.1)	63	(11.4)
	Stage 4 or 5	54	(100.0)	22	(40.7)	13	(24.1)	8	(14.8)	4	(7.4)	7	(13.0)

SSB: sugar-sweetened beverages.

**Table 2 nutrients-14-02720-t002:** The mortality risk of cardiovascular diseases by levels of sugar-sweetened beverages consumed in younger adults.

SSB Intake (Serving/Day)	0–<0.5		≥0.5–1			≥1–<2			≥2			*p* for Trend	≥1		
Causes of Deaths	N	HR	N	HR	95% CI	N	HR	95% CI	N	HR	95% CI		N	HR	95% CI
CVD	242	Ref	47	1.03	0.73–1.45	48	1.14	0.82–1.59	54	1.59	1.16–2.17	0.009	102	1.34	1.04–1.73
Ischemic heart disease	53	Ref	3	0.33	0.10–1.07	9	0.90	0.42–1.94	14	1.78	0.95–3.32	0.200	23	1.31	0.77–2.23
Stroke	72	Ref	17	1.24	0.70–2.21	13	1.14	0.62–2.10	16	1.51	0.83–2.74	0.196	29	1.30	0.82–2.09
Diabetes mellitus	31	Ref	5	1.26	0.47–3.38	5	0.99	0.34–2.89	5	1.20	0.45–3.21	0.749	10	1.10	0.50–2.40
Kidney diseases	10	Ref	0			4	2.38	0.67–8.50	0			0.745	4	1.18	0.33–4.20
Expanded CVD	283	Ref	52	1.02	0.74–1.40	57	1.18	0.87–1.60	59	1.49	1.11–2.01	0.011	116	1.32	1.04–1.68

HR adjusted for age, sex, education levels, smoking status, alcohol drinking status, physical activity, body mass index, hypertension, and diabetes. SSB: sugar-sweetened beverages.

**Table 3 nutrients-14-02720-t003:** Sensitivity analyses for the mortality risks of cardiovascular diseases by levels of sugar-sweetened beverages consumed.

SSB Intake (Serving/Day)	0–<0.5		≥0.5–<1			≥1–<2			≥2			P for Trend	≥1		
	N	HR	N	HR	95% CI	N	HR	95% CI	N	HR	95% CI		N	HR	95% CI
Men	170	Ref.	34	0.95	0.64–1.42	36	1.16	0.79–1.70	41	1.48	1.03–2.12	0.046	77	1.31	0.98–1.76
Women	72	Ref.	13	1.27	0.68–2.39	12	1.05	0.55–2.02	13	1.90	1.01–3.61	0.109	25	1.37	0.83–2.25
Non-smokers	94	Ref.	22	1.22	0.75–1.99	22	1.28	0.77–2.12	17	1.95	1.14–3.32	0.020	39	1.52	1.02–2.28
Ever smokers	133	Ref.	25	0.89	0.56–1.43	26	1.04	0.67–1.61	36	1.42	0.96–2.09	0.134	62	1.23	0.89–1.70
BMI < 30 kg/m^2^	206	Ref.	37	0.97	0.66–1.42	44	1.22	0.86–1.73	43	1.54	1.08–2.19	0.019	87	1.36	1.03–1.79
BMI ≥ 30 kg/m^2^	36	Ref.	10	1.40	0.64–3.03	4	0.69	0.24–1.99	11	1.85	0.89–3.81	0.249	15	1.27	0.66–2.44
Non-hypertension	156	Ref.	26	0.87	0.55–1.38	37	1.40	0.96–2.06	40	1.89	1.30–2.73	0.001	77	1.62	1.20–2.18
Hypertension	86	Ref.	21	1.31	0.79–2.17	11	0.67	0.33–1.35	14	1.06	0.58–1.94	0.797	25	0.85	0.52–1.39
Non-diabetes	223	Ref	45	1.04	0.73–1.47	45	1.15	0.82–1.62	49	1.54	1.11–2.14	0.017	94	1.33	1.02–1.73
Diabetes	19	Ref.	2	0.81	0.18–3.74	3	0.90	0.19–4.18	5	2.20	0.76–6.38	0.247	8	1.58	0.61–4.10
Daily calorie intake ≥25 Kcal/Kg body weight	60	Ref	20	1.03	0.60–1.78	29	1.40	0.86–2.26	36	1.87	1.19–2.95	0.006	65	1.78	1.09–2.91
Daily calorie intake <25 Kcal/Kg body weight	165	Ref	24	1.14	0.72–1.80	15	0.88	0.51–1.55	17	1.75	1.04–2.93	0.160	32	1.31	0.85–2.01
Exclude those died in 3 years	222	Ref.	42	1.07	0.75–1.53	41	1.08	0.75–1.54	51	1.66	1.20–2.29	0.010	92	1.34	1.03–1.75
Exclude those with smoking, hypertension, or diabetes	57	Ref	11	0.96	0.48–1.92	17	1.63	0.90–2.93	13	2.48	1.33–4.62	0.004	30	1.93	1.20–3.12
Exclude regular alcohol drinkers	242	Ref.	47	1.03	0.73–1.45	48	1.14	0.82–1.59	54	1.58	1.16–2.17	0.009	102	1.34	1.04–1.73

Except for the stratification criteria, HR was adjusted for categories of age, sex, education levels, smoking status, alcohol drinking status, physical activity, body mass index, hypertension, and diabetes. SSB: sugar-sweetened beverages.

**Table 4 nutrients-14-02720-t004:** Amount of sugar-sweetened beverage intake for participants with two visits.

		First Visit (n)					
	SSB Intake (Serving/Day)	None	>0–<0.5	≥0.5–<1	≥1–<2	≥2	Total
Second visit (n)	None	14,391	7783	2344	1712	931	27,161
	>0–<0.5	9541	15885	6959	4370	1836	38,591
	≥0.5–<1	3314	5798	5895	4160	1734	20,901
	≥1–2	2818	3194	3582	5615	2784	17,993
	≥2	1676	1164	1094	1862	3539	9335
	Total	31,740	33,824	19,874	17,719	10,824	

SSB: sugar-sweetened beverages.

## Data Availability

The data used in this research were authorized by the MJ Health Research Foundation (Authorization Code: MJHRFB2014001C). The MJ Health Research Foundation administered MJ Health Survey Database and MJ BioData, and the data were available at the following website: http://www.mjhrf.org (accessed on 31 May 2022). The interpretation and conclusion in this paper do not represent the viewpoints of the MJ Health Research Foundation.

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
