# Peer review of "Association of Sugar-Sweetened Beverages and Cardiovascular Diseases Mortality in a Large Young Cohort of Nearly 300,000 Adults (Age 20–39)"

_nutrients, 2022, doi:10.3390/nu14132720_

Round 1
Reviewer 1 Report
Thanks to the editor for the invitation. In this study, the authors have investigated the association between SSB consumption and risk of CVD mortality in young adults(?). The advantages of this study including the large sample size and long-term of follow-up. Please see my comments below.
1, The title of this study indicated the age range of participants was 20-29 years old, whereas the age range of participants was 20-39 years old in the method section. Please clarify it. If the age range is 20-39 years old, the implications and significance of this study will be decreased greatly.
2, Actually, there are a lot of previous studies have investigated the SSB consumption and risk of CVD mortality by age groups.
3, The SSB consumption will decrease with age, the lack of SSB consumption during the follow-up period is a major limitation in this study. The change of SSB consumption is also associated with weight change and CVD.
4, Stratified analyses by sex, smoking, diabetes, BMI and other confounders are needed.
Author Response
Dear Editors, June 24, 2022
Many thanks for your attention to our paper again. We are submitting the revised manuscript entitled “Sugar-sweetened beverages and mortality from cardiovascular diseases for the younger adults (age 20-39): Based on 288,747 Taiwanese cohort followed for 25 years” (Manuscript ID: nutrients-1775539) for your consideration. We have revised this paper point by point according to the reviewers’ comments, and the revised contents were marked by red fonts. Our response to the reviewers’ comments would also be provided as the followings.
All authors have read and approved the submitted manuscript, this paper has not been submitted elsewhere nor published elsewhere in whole or in part. We believe it will be interesting and educative to your readers if published.
Sincerely,
Chi Pang Wen, MD DrPH (Harvard)
Distinguished Professor
National Health Research Institutes
Zhunan, Taiwan

Reviewer 2 Report
Dear Editor,
I carefully read the manuscript by Chen et al.
My comments and suggestions are the following:
- English language needs to be carefully revised and improved.
- The manuscript is not balanced in its parts. In particular, the Introduction should be shortened.
- "Based on Encyclopedia Britannica, the definition of the younger adults was made for the adults aged between 20 and 39". I don't understand why this sentence was included in the Introduction instead of the Methods.
- "Gender" should be replaced by "sex". In effect, sex tends now to refer to biological differences, while gender often refers to cultural or social ones.
- Some references need to be updated.
- The authors should consider to refer to doi: 10.3390/nu11112674 in their manuscript.
Author Response
#Reviewer 2
Dear Editor,
I carefully read the manuscript by Chen et al.
My comments and suggestions are the following:
- English language needs to be carefully revised and improved.
Ans: Many thanks for the reviewer’s comment and the English language has been corrected by a native English speaker with medical background.
- The manuscript is not balanced in its parts. In particular, the Introduction should be shortened.
Ans: The section of Introduction has been shortened accordingly.
- "Based on Encyclopedia Britannica, the definition of the younger adults was made for the adults aged between 20 and 39". I don't understand why this sentence was included in the Introduction instead of the Methods.
Ans: This statement has been moved to the section of Materials and Methods.
- "Gender" should be replaced by "sex". In effect, sex tends now to refer to biological differences, while gender often refers to cultural or social ones.
Ans: The wording “gender” has been replaced by “sex”.
- Some references need to be updated.
Ans: Many thanks for the reviewer’s comment and we have extensively updated the references accordingly.
- The authors should consider to refer to doi: 10.3390/nu11112674 in their manuscript.
Ans: We have referred to this reference in #29 reference.
Round 2
Reviewer 1 Report
The response looks good, whereas investigating the SSB consumption during childhood and adolescents and risk of CVD in young adulthood may be an interesting one.
Reviewer 2 Report
Dear Editor,
I carefully read the revised version of the manuscript that is significantly improved in comparison with the previous one. I warmly recommed its publication in the Journal.